# Sedoheptulose Kinase SHPK Expression in Glioblastoma: Emerging Role of the Nonoxidative Pentose Phosphate Pathway in Tumor Proliferation

**DOI:** 10.3390/ijms23115978

**Published:** 2022-05-26

**Authors:** Sara Franceschi, Francesca Lessi, Mariangela Morelli, Michele Menicagli, Francesco Pasqualetti, Paolo Aretini, Chiara Maria Mazzanti

**Affiliations:** 1Fondazione Pisana per la Scienza, 56017 Pisa, Italy; f.lessi@fpscience.it (F.L.); m.morelli@fpscience.it (M.M.); m.menicagli@fpscience.it (M.M.); p.aretini@fpscience.it (P.A.); c.mazzanti@fpscience.it (C.M.M.); 2Department of Radiation Oncology, Azienda Ospedaliera Universitaria Pisana, University of Pisa, 56126 Pisa, Italy; f.pasqualetti@ao-pisa.toscana.it; 3Department of Oncology, University of Oxford, Oxford OX3 7DQ, UK

**Keywords:** glioblastoma, cancer metabolism, pentose phosphate pathway, cell proliferation

## Abstract

Glioblastoma (GBM) is the most common form of malignant brain cancer and is considered the deadliest human cancer. Because of poor outcomes in this disease, there is an urgent need for progress in understanding the molecular mechanisms of GBM therapeutic resistance, as well as novel and innovative therapies for cancer prevention and treatment. The pentose phosphate pathway (PPP) is a metabolic pathway complementary to glycolysis, and several PPP enzymes have already been demonstrated as potential targets in cancer therapy. In this work, we aimed to evaluate the role of sedoheptulose kinase (SHPK), a key regulator of carbon flux that catalyzes the phosphorylation of sedoheptulose in the nonoxidative arm of the PPP. SHPK expression was investigated in patients with GBM using microarray data. SHPK was also overexpressed in GBM cells, and functional studies were conducted. SHPK expression in GBM shows a significant correlation with histology, prognosis, and survival. In particular, its increased expression is associated with a worse prognosis. Furthermore, its overexpression in GBM cells confirms an increase in cell proliferation. This work highlights for the first time the importance of SHPK in GBM for tumor progression and proposes this enzyme and the nonoxidative PPP as possible therapeutic targets.

## 1. Introduction

Glioblastoma (GBM) is one of the most aggressive and deadly types of central nervous system tumors [1,2]. GBMs make their way into surrounding brain tissue in widespread and unpredictable ways, making surgical resection a nearly impossible mission [3,4,5]. The rapid evolution and progression of the tumor and the extreme heterogeneity of GBM, even within the same tumor, means that most currently available cancer treatments fail to be effective [6,7]. Survival rates from GBM enjoyed a modest bump in the 1980s when radiation became a standard part of the treatment protocol [8,9]. Patients could expect to live for almost another year after diagnosis, from just four to six months. The introduction of the chemotherapy drug temozolomide in the 2000s increased survival by another few months [10]. However, since then, patient survival rates have stalled. Because of the poor outcomes of this disease, primarily due to its daunting resistance to almost all forms of treatment, new and innovative therapies are urgently needed. Impairment of physiological metabolism is one of the most striking hallmarks of GBM [11,12], and the cancer cells’ propensity to use glycolysis instead of oxidative phosphorylation (Warburg effect) [13,14] led to a focus on glucose metabolism to stop tumor progression. In addition to glycolysis, which is considered the main energy source in tumors, other metabolic pathways are specifically exploited in GBM [13] to meet the demand of a rapidly proliferating tumor. The pentose phosphate pathway (PPP) is a metabolic pathway parallel to glycolysis and represents the first committed step of glucose metabolism [15]. The PPP plays a critical role in sustaining cancer cell survival and growth by producing ribose-5-phosphate (R5P) for nucleic acid synthesis and providing nicotinamide adenine dinucleotide phosphate (NADPH), which is necessary for fatty acid synthesis and cell survival under high-stress conditions [16,17]. Indeed, NADPH and R5P play critical roles in the regulation of metabolism, proliferation, and DNA damage response in cancer cells, and several PPP enzymes have already been studied, highlighting their potential role as molecular targets for the development of cancer therapies [18]. In GBM in particular, most studies have investigated the enzyme glucose-6-phosphate dehydrogenase (G6PD), which is part of the oxidative arm of the PPP, demonstrating a prognostic relevance [19]. Although the role of the nonoxidative arm of the PPP has been little studied in GBM, its enzymes and the Sedo-heptulose-7-phosphate (S7P) intermediate appear to play an important role in other malignancies [20,21]. In this work, we aimed to evaluate the role of sedoheptulose kinase (SHPK), a key regulator of carbon flux that catalyzes the phosphorylation of sedoheptulose into S7P, making it available to cells. As S7P is the substrate of two enzymes in the nonoxidative arm, Transketolase (TKT) and Transaldolase (TALDO1), the action of SHPK is important in regulating flux through the PPP. SHPK expression in GBM shows a significant correlation with histology, patient prognosis, and survival. Moreover, its overexpression in GBM cells confirms an increase in tumor proliferation.

## 2. Results

### 2.1. Correlation between SHPK Expression and Clinical Characteristics of Glioma Patients

Clinical and gene expression data from 219 GBMs, 225 low-grade tumors, primary tumors, and 28 normal specimens were used to correlate SHPK mRNA expression levels with histopathological features through the Rembrandt dataset [22] via GlioVis, a web-based data visualization and analysis application for exploring brain tumor expression datasets [23].

Results show a significantly positive correlation between histology and WHO grade and SHPK expression (Figure 1A,B). Using 219 GBMs, we further investigated how SHPK mRNA expression correlates with a specific molecular subtype (defined by Wang [24]) and patient survival. As shown in Figure 1C, SHPK was significantly upregulated in the classical and mesenchymal subtypes compared with the proneural subtype. To evaluate the prognostic value of SHPK in GBM samples, Kaplan–Meier (KM) survival curves were plotted. We observed that higher SHPK mRNA expression predicted a significantly shorter survival, as shown in Figure 1E.

Moreover, we verified the protein levels of SHPK in glioma tissues using the Human Protein Atlas (HPA) database (available from http://www.proteinatlas.org, accessed on 10 January 2022). The immunohistochemistry data in the HPA database reveal that the immunoreactive score (IRS) of SHPK was significantly higher in glioma tissues (both low- and high-grade gliomas) than in the normal cerebral cortex tissues (Figure 1D).

### 2.2. SHPK-Related Biological Process

Expression data of the 219 GBMs were employed to conduct a differential expression analysis. In total, 263 differentially expressed genes (DEGs) were identified by dividing GBM samples according to their SHPK expression (high vs. low: split into two groups, 25% lower-expressing vs. 25% higher-expressing). To better understand the function of the identified DEGs, GO analysis was performed in g:Profiler [25] as g:GOSt functional profiling. Table 1 shows the results of the functional enrichment analysis of g:Profiler exploiting the Reactome [26] database.

GBM samples were divided into two groups according to their SHPK mRNA expression (high and low). The top table shows the biological pathways in which genes overexpressed in samples with the lowest SHPK mRNA values (25% lower-expressing) are involved, while the bottom shows the biological pathways of genes overexpressed in samples with the highest SHPK mRNA values (25% higher-expressing). Reactome, biological pathway description; ID, biological pathway identifier (REAC:R-HSA); *p*-value, adjusted enrichment *p*-values in negative log10 scale; Black square indicates which gene belongs to that pathway.

The biological processes most represented by genes overexpressed in GBMs with low SHPK mRNA values are nervous system development, synaptic signaling, cell–cell signaling, and neurogenesis. The most significant biological processes to which overexpressed genes in GBMs with high SHPK mRNA expression belong are extracellular matrix and structure organization, response to endogenous stimuli, and regulation of cell population proliferation.

### 2.3. SHPK Correlation with Other PPP Enzymes

We then conducted a correlation analysis with the expression data from the 219 GBMs. In particular, we conducted Pearson correlation analysis of the mRNA expression of SHPK with the mRNA expression of the other enzymes constituting the PPP (Figure 2A,B). Specifically, Figure 2A shows the correlations performed among the enzymes in the nonoxidative arm of the PPP while Figure 2B shows the correlations performed among the enzymes comprising the oxidative arm of the PPP. SHPK correlated significantly and positively with Ribose 5-Phosphate Isomerase A (RPIA, nonoxidative branch of PPP) and with 6-Phosphogluconolactonase, Phosphogluconate Dehydrogenase, and Hexose-6-Phosphate Dehydrogenase/Glucose 1-Dehydrogenase (PGLS, PGD, and H6PD, oxidative branch of PPP).

### 2.4. SHPK Expression and Mutational Status Association

The LinkFinder module of LinkedOmics [27] was used to conduct multi-omics analyses within another cohort of 595 GBM patients. We conducted an association analysis between mutational status (whole-exome data) and SHPK mRNA expression levels (RNAseq data). Association analysis results are shown in Figure 2C (volcano plot). SHPK mRNA expression levels were assessed for each gene for its mutational status (mutated or WT) with the Wilcoxon test. The mutational status of nine genes, Isocitrate Dehydrogenase 1 (IDH1), Tumor Protein P53 (TP53), ATRX Chromatin Remodeler (ATRX), Sodium Voltage-Gated Channel Alpha Subunit 9 (SCN9A), Phosphoinositide-3-Kinase Regulatory Subunit 1 (PIK3R1), Frizzled Class Receptor 10 (FZD10), Polycystic Kidney Additionally, Hepatic Disease 1 (Autosomal Recessive)-Like 1 (PKHD1L1), Armadillo Repeat-Containing 3 (ARMC3), and Acyl-CoA Synthetase Medium Chain Family Member 2B (ACSM2B), was significantly associated with a lower SHPK mRNA expression (Figure 2C and Table 2). In contrast, 13 other genes’ mutated state was significantly associated with increased SHPK mRNA expression (Figure 2C and Table 2). These were Epidermal Growth Factor Receptor (EGFR), Vacuolar protein sorting-associated protein 8 homolog (VPS8), Rho Guanine Nucleotide Exchange Factor 16 (ARHGEF16), Striated-Muscle-Enriched Protein Kinase (SPEG), Cadherin 9 (CDH9), Transformation/Transcription-Domain-Associated Protein (TRRAP), Ryanodine Receptor 2 (RYR2), Solute Carrier Family 4 Member 1 (SLC4A1), Molybdenum Cofactor Synthesis 3 (MOCS3), Dynein Axonemal Heavy Chain 2 (DNAH2), Lysine Methyltransferase 2D (MLL2), Xin-Actin-Binding Repeat-Containing 2 (XIRP2), and APC Membrane Recruitment Protein 3 (FAM123C).

**Figure 2 ijms-23-05978-f002:**
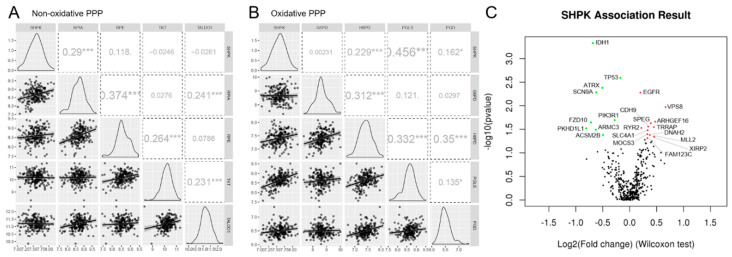
SHPK correlation and association analysis. Pearson correlation analysis between mRNA expression of enzymes constituting the nonoxidative (**A**) and oxidative (**B**) arms of PPP. The figure shows the scatter plot with regression line for each correlation (bottom diagonal), the density plot (middle diagonal), and the Pearson correlation coefficient with significance: *** *p* < 0.001, * *p* < 0.05 (upper diagonal). (**C**) Volcano plot showing the log2 (fold change) vs. log10 (*p*-value) obtained from the analysis. In green, with a negative Log2FC, statistically significant associations between the absence of mutation and SHPK mRNA overexpression and between the presence of mutation and downregulation of SHPK mRNA expression are highlighted. In red, with a positive Log2FC, statistically significant associations between the presence of the mutation and SHPK mRNA overexpression and between the absence of the mutation and downregulation of SHPK mRNA expression are highlighted.

### 2.5. SHPK Overexpression and Cell Functional Studies

SHPK has been successfully cloned and overexpressed in three commercial human GBM lines (T98G, U118, and U87) with an average transfection efficiency of 68% for the vector and 59% for the vector containing SHPK (data not shown). Both SHPK mRNA and protein expression were significantly higher in SHPK-overexpressed cells than in vector cells (Figure 3A,B).

The WST-1 cell proliferation test was used to determine the effect of SHPK overexpression in T98G, U118, and U87 cells at four different time points, T0, T1 (24 h), T2 (48 h), and T3 (72 h). It was found that SHPK overexpression significantly enhanced cell proliferation (Figure 3C). The effect of SHPK overexpression on cell migration was studied with the wound healing assay. In all three cell lines (U118, U87, and T98G), four different time points were taken, T0, T1 (24 h), T2 (48 h), and T3 (72 h). Wound healing test results showed that SHPK overexpression did not change the wound healing ability compared to transfected empty vector cells (Figure 3D). To determine the effect of SHPK overexpression on T98G, U118, and U87 cell invasion, a transwell invasion assay was performed. The number of migrating cells was unchanged in SHPK-overexpressed cells compared to control cells (Figure 3E). Colony formation assays were performed to evaluate the SHPK overexpression effect on clonogenic survival. SHPK overexpression did not change the T98G, U118, or U87 colony formation ability (data not shown). To gain insight into metabolic differences in cells after SHPK overexpression, we analyzed metabolic phenotypes and the metabolic potential of live cells with the Seahorse XFp extracellular flow analyzer. We examined the effects of SHPK overexpression on the oxygen consumption rate (OCR), which is a measure of the rate of mitochondrial respiration of cells, and the extracellular acidification rate (ECAR), a measure of the rate of glycolysis of cells using the cell energy phenotype assay (Figure 3F). Although there were differences in basal metabolic status due to SHPK overexpression, this did not occur equally in the three cell lines (Figure 3F). In particular, the basal metabolic status appeared to shift toward glycolysis when SHPK was overexpressed in T98G (Figure 3F). In U118 and U87, however, it shifted to lower values of glycolytic activity; even in U87, it went toward a more quiescent metabolic state (Figure 3F). Regarding the metabolic differences of the stressed phenotype (cells under an induced energy demand), we can observe in Figure 3F that stress increased mitochondrial respiration in all three cell lines (OCR) independently of SHPK overexpression, whereas cell glycolytic activity (ECAR) appeared to increase under stress in all lines and under both SHPK expression conditions except for U118 cells without overexpression (Figure 3F).

## 3. Discussion

GBM is the most common and malignant primary brain tumor, characterized by high morbidity and poor survival [28]. Despite advances in diagnosis and treatment, life expectancy remains at approximately 12–18 months [1,29]. Brain invasion, motility, and rapid proliferation are characteristic of GBM cells, and this ability to invade surrounding tissue is a major determinant for malignant tumor progression [30,31]. The dispersion of tumor cells from the primary tumor site into adjacent brain tissue results in rapid and almost inevitable recurrence [29,32].

GBM cells are characterized by a preference for aerobic glycolysis rather than oxidative phosphorylation, which is more active in normal cells [33,34]. However, in addition to elevated glycolysis, proliferating and cancer cells must also divert carbon from glycolysis to the pentose phosphate pathway (PPP) to satisfy their anabolic demands and maintain the redox homeostasis of cells [21]. The PPP plays a key role in the regulation of cancer cell growth by producing ribose-5-phosphate and NADPH for detoxification of intracellular ROS, reductive biosynthesis, and ribose biogenesis [35]. Thus, the PPP is directly related to cell proliferation, survival, and senescence. It has been reported that in GBM cells, migrating cells are characterized by up-regulation of many key glycolysis enzymes at the expense of PPP enzyme expression, whereas in rapidly dividing GBM cells, the opposite occurs: PPP enzyme expression increases, and glycolysis enzymes decrease their expression [16,35,36]. The PPP is mainly used during proliferation, and glycolysis is used as the energy source during migration [37]. The metabolism acts as a mutual switch between the two pathways, glycolysis, and the PPP, and the mechanisms of cancer cell invasion and proliferation are thought to be mutually exclusive behaviors, called the “migration–proliferation dichotomy” or “go or grow” [16,35,38].

The PPP is composed of two functionally interrelated branches: the oxidative and the nonoxidative. The oxidative branch consists of three irreversible reactions leading to the generation of NADPH and ribonucleotides [15,21]. Additional glycolytic intermediates such as fructose-6-phosphate (F6P) and glyceraldehyde-3-phosphate (G3P) are recruited into the nonoxidative branch through a series of reversible reactions [18,21,39]. This reversible nature of the nonoxidative branch of the PPP makes it capable of adapting to the metabolic demands of cells, acting in a multitude of ways [21,40].

Previous studies have examined the relative gene expression of enzymes involved in the oxidative phase of the PPP, finding an overall increase in glycolytic and PPP genes driving ATP production and excess nucleotides resulting in uncontrolled proliferation in GBM cells [36,41,42]. In addition, inhibitors of the oxidative branch of the PPP have been studied, the findings of which indicate increased radiosensitivity in human gliomas [43]. Although the oxidative branch of the PPP has already been studied in GBM by correlating enzyme expression with tumor aggressiveness and patient survival and proposing their targeting as a promising therapeutic target, the nonoxidative branch of the PPP has never been sufficiently evaluated in GBM. However, there are studies on other cancer types that confer an important role in tumor progression on the nonoxidative branch of the PPP and overexpression of its enzymes mainly due to increased proliferation of cancer cells [15,21,34,44,45,46].

In this work, we focused on the sedoheptulokinase (SHPK) protein belonging to the nonoxidative branch enzymes of the PPP. SHPK is the carbohydrate kinase that catalyzes the phosphorylation of sedoheptulose into sedoheptulose 7-phosphate (S7P), which then enters the PPP stream [18]. S7P represents a glycolysis-independent entry and exit point into/out of nonoxidative PPP [40] and has been shown to play an important role in other malignancies [20,21,47] and found in greater amounts in high-grade than low-grade gliomas [48]. Since studies suggest that a role of SHPK might be to provide increased PPP flow during an increased need for energy [40], we investigated the expression of this enzyme in GBM. With this work, we want to draw attention to the importance of the increased production of S7P by SHPK by correlating the expression of this enzyme with progression and tumor aggressiveness in GBM. We initially evaluated how SHPK expression correlated with the clinical characteristics of patients. SHPK mRNA expression was significantly higher in tumor tissues than in nontumor tissues and among different histologies. In particular, SHPK was highest in GBM, then in oligodendrogliomas, and significantly lower in astrocytomas. Within the adult-type diffuse gliomas (WHO-grade II-IV [49]) there was a significant difference between GBM (grade IV), in which SHPK was overexpressed, and grade II. In addition, a significant overexpression of SHPK protein in gliomas compared with normal tissues was found. The expression of SHPK within GBM samples was evaluated among the three different molecular subtypes defined by Wang [24]. Proneural GBMs show a significant decrease in SHPK mRNA expression. Compared to the other three subtypes, proneural subtypes have better survival rates [50]. This finding was also confirmed by Kaplan–Meier survival analysis, showing a significant difference between the survival of GBM patients characterized by high SHPK mRNA levels compared to those with low SHPK mRNA expression.

To understand which cellular pathways were related to the different expressions of SHPK in GBMs and what this increased flux in the PPP of the S7P intermediate led to, we conducted a functional enrichment analysis. In agreement with that described in the literature [16,21,35,36,37,41,42], increased SHPK mRNA expression and a subsequent S7P flux in the PPP triggered a number of pathways involved in cell proliferation. In contrast, GBMs characterized by a low level of SHPK mRNA showed active physiological cell signaling pathways and had no differentially expressed genes involved in extracellular matrix remodeling and proliferation pathways. Next, we also assessed how SHPK mRNA expression correlated with the expression of the other PPP enzymes in GBM, both the nonoxidative and oxidative branches. Significant correlations of SHPK with other enzymes in the PPP were all positive. In the nonoxidative branch, only one enzyme correlated with SHPK (RPIA), while in the oxidative branch, three enzymes positively correlated with SHPK (PGLS, PGD, and H6PD). This may suggest that the two pathways do not operate separately but rather that they may work in synergy, or one may compensate for the reduced work of the other. Metabolic control analyses performed on PPP regulatory enzymes have been carried out previously, which revealed that the nonoxidative branch of the PPP is more important for tumor growth than the oxidative one [51]. On the other hand, the results obtained in other studies [52,53] demonstrate the importance of a forced balance of the activity of the two branches in the direction of the oxidative one to sustain high tumor cell proliferation. When dividing the GBM population into high and low SHPK expression, we could observe differences in the mutational status of some important genes in GBM. For example, the most significant genes in each group were IDH1, which associates its mutated state with lower SHPK expression, and EGFR, which instead associates its mutated state with higher SHPK expression. This agrees perfectly with what we found in the literature, where strong evidence shows that IDH1 mutation is associated with a better prognosis for GBM patients [54,55,56], and on the other hand, EGFR mutation with a worse one [57].

Finally, we evaluated the effects of SHPK overexpression in three different GBM cell lines (T98G, U87, and U118) to try to associate a particular cell behavior with increased SHPK and the respective intermediate of the S7P PPP. In all three cell lines, the increase in SHPK alone was able to enhance their viability/proliferation. On the other hand, no other cellular capacity (invasion, migration, and clonogenicity) was altered by increasing SHPK. This finding further confirms the key role of the nonoxidative PPP in tumor proliferation but especially highlights and proposes a possible leading role of SHPK in the activation of the nonoxidative branch. To conclude the functional studies, we evaluated metabolic phenotypes and the metabolic potential of GBM lines after SHPK overexpression. With this type of assay, we were able to observe whether SHPK expression can affect the metabolic state of GBM cells both before (basal) and after an induced energy demand. Although there were differences, we could not find one that was reflected in all three cell lines. This is most likely because SHPK alone is unable to trigger specific metabolic switches. However, it should be kept in mind that these assays were conducted within 48 days of transfection with one day of settling. Surely real-time studies would be helpful to better understand if there are indeed metabolic differences following SHPK overexpression, and permanent transfection studies could be more informative.

In conclusion, we can state that expression and therefore the activity of SHPK to produce S7P that enters the PPP stream could trigger the activity of the nonoxidative branch. This results in an increase in cell proliferation that, through functional studies, has been attributed to the activity of SHPK alone. Moreover, SHPK expression is also significantly correlated with multiple clinical data proposing the correlation of its expression with a worse prognosis. This work highlights for the first time in GBM the importance of the SHPK enzyme, although the mechanisms and flow direction by which the PPP is activated and which enzymes are primarily involved are still unclear. Further studies are needed to better understand the flux of metabolites through the nonoxidative PPP and in particular to understand how we can exploit the SHPK enzyme as a therapeutic target.

## 4. Materials and Methods

### 4.1. Glioma Samples and Normal Controls

Clinical and gene expression data from 219 GBMs, 225 low-grade tumors, primary tumors, and 28 normal samples were analyzed using microarray data from the Rembrandt cohort through the data visualization and analysis tool GLIOVIS [23] (http://GLIOVIS.bioinfo.cnio.es/, accessed on 11 January 2021) to correlate SHPK expression with histopathological characteristics. Correlation of SHPK with molecular subtype (classic, mesenchymal, and proneural), and survival was conducted considering only the 219 GBMs. Multiomics analysis to associate gene mutation states with SHPK mRNA expression in GBM was conducted on the TCGA (The Cancer Genome Atlas) cohort of 595 samples, using the LinkFinder module of LinkedOmics [27]. We selected RNAseq data as the select search (query) dataset (Illumina HiSeq 2000 RNA Sequencing) and SHPK as the gene of interest. Mutation datatype (Illumina GA IIx) was chosen as the target dataset. T-test was chosen as the statistical analysis method.

### 4.2. Immunohistochemistry of Histological Sections

Information used for immunohistochemistry and annotation data was provided by Human Protein Atlas (proteinatlas.org, accessed on 10 January 2022). Basic annotation parameters included an assessment of staining intensity (negative, weak, moderate, or strong) and a fraction of stained cells (<25%, 25–75%, or >75%). As shown in Table 3, each tumor was assigned a score based on staining intensity (no staining = 0; weak staining = 1; moderate staining = 2; strong staining = 3) and extent of stained cells (0% = 0; <25% = 1; 25–75% = 2; >75% = 3). The final immunoreactivity score (IRS) was determined by multiplying the intensity and extent of stained-cell positivity scores, with a minimum score of 0 and a maximum score of 9. In gliomas, tumor cell staining was considered, whereas in healthy tissues, glial cell staining was considered.

### 4.3. Cell Lines and Transfection

T98G, U87, and U118 GBM cell lines were obtained from American Type Culture Collection (ATCC, Rockville, MD, USA). To ensure the quality and integrity of the human cell lines, STR analysis was conducted using the GenePrint 10 system (Promega, Madison, WI, USA). Cells were grown as monolayers in high-glucose Dulbecco’s Modified Eagle Medium (DMEM) supplemented with 10% FBS and 1% penicillin–streptomycin. Cells were tested for the presence of mycoplasma (EZ-PCR Mycoplasma Test Kit; Biological Industries, Kibbutz Beit-Haemek, Israel) with negative results. SHPK was overexpressed using the pCMV6-AC-GFP vector with the molecular sequence of its clone (NM_013276) (OriGene Technologies, Rockville, MD, USA) cloned within, as a control. The SHPK sequence was confirmed by Sanger sequencing. Transfection of the plasmid was performed with Lipofectamine 3000 reagent, following the manufacturer’s instructions. Cells were incubated for 48 h after SHPK overexpression before characterization and functional experiments.

### 4.4. SHPK mRNA Expression of Cell Lines

Total cellular RNA was extracted from GBM cells using the Maxwell 16 LEV simplyRNA kit (Madison, WI, USA) according to the manufacturer’s instructions, and quantified using a Qubit 2.0 Fluorometer (Thermo Fisher Scientific, Waltham, MA, USA). Total RNA was inversely transcribed into cDNA using the RT-NanoScript kit (PrimerDesign, Southampton, UK). Real-time PCR was performed following the manufacturer’s instructions for the SsoAdvanced SYBR Green Supermix kit (Bio-Rad, Hercules, CA, USA) on the CFX96 instrument (Bio-Rad). TBP expression values were used for normalization. Real-time PCR primer assays (Bio-Rad) for SHPK (Assay ID: qHsaCID0016666) and TBP (Assay ID: qHsaCID0007122) were used. Gene expression analysis was performed using CFX Manager software (Bio-Rad). All expression experiments were performed in triplicate.

### 4.5. Immunofluorescence

Cells were grown on cell culture chamber slides and fixed in 1.5% paraformaldehyde for 15 min. Cells were permeabilized with 0.1% Triton X-100 for 15 min and blocked with 2% BSA for 45 min. Primary SHPK antibody (HPA024361, Sigma Aldrich, St. Louis, MO, USA) was diluted at 1:50 and incubated for 60 min at RT. Phycoerythrin-conjugated secondary antibody (P9287, Sigma Aldrich) was diluted at 1:20 and incubated for 30 min. Cells were counterstained with DAPI (Thermo Fisher Scientific) and visualized using the CARL ZEISS Axio Observer 3 Z1FLMot inverted microscope (Zeiss, Gina, Germany).

### 4.6. Cell Viability Assay

Cell viability was determined using the WST1 assay (Clontech Laboratories, Mountain View, CA, USA). A total of 5000 cells per well were seeded in a 96-well plate. At the time of seeding (T0) and after 24 h (T1), 48 h (T2), and 72 h (T3), the WST1 reagent was added and incubated for an additional 60 min before reading the plate. Each assay was conducted in triplicate. The amount of formazan dye was directly related to the number of metabolically active cells and was quantified by measuring absorbance at 450 nm in a multiwell plate reader (Tecan, Mannedorf, Switzerland). OD values at 24 h (T1), 48 h (T2), and 72 h (T3) were normalized to T0.

### 4.7. Wound Healing Assay

Cells were plated in Culture-Insert 2 Well in 35 mm μ-Dish (IBIDI, Martinsried, Germany) until cells were confluent or nearly confluent (>90%). After removal of the insert, cell migration in the wound area was observed and digitally photographed. Wound healing was measured on the images using the free and open-source ImageJ software [58], and % closure was calculated at each time (T0–T3, 0–72 h) as the area to be healed divided by the area of the original wound × 100. Experiments were performed in triplicate.

### 4.8. Transwell Assay

Cell invasion was assessed using 24-well inserts (Sarstedt, Nuembrecht, Germany) with 5 μm pores according to the manufacturer’s instructions. Briefly, 1 × 10^5^ cells were seeded in the upper chamber with 1% FBS medium and were allowed to invade the lower reservoir, containing 10% FBS, at 37 °C for 24 h. Noninvasive cells in the upper surface of the filters were removed with a cotton swab. The remaining cells were fixed in 70% ethanol and stained with 0.01% crystal violet for 30 min. Cells that crossed the membrane were counted in five visual fields as migrated cells. The experiment was performed in triplicate.

### 4.9. Clonogenic Survival Assay

Cells were seeded at 500 cells/well in 6-well plates and incubated for 2 weeks. Cells were fixed with 70% ethanol and stained with 0.01% crystal violet for 30 min. The mean ± SD number of colonies >50 μm in diameter was counted microscopically in five nonoverlapping fields in three independent experiments.

### 4.10. Cell Energy Phenotype Test

Cell mitochondrial function was evaluated by using the Seahorse XFp Cell Energy Phenotype Test Kit on the Seahorse XFp Analyzer (Agilent Technologies, Santa Clara, CA, USA). Cells were seeded at 20,000 cells per well into XFp well cell culture plates and incubated overnight at 37 °C in a 5% CO_2_-humidified atmosphere in Seahorse XF Base Medium (Agilent Technologies) with 1 mM pyruvate, 2 mM glutamine, and 10 mM glucose. Cartridge compounds were loaded to obtain a final concentration of 1 μM Oligomycin and 1 μM FCCP. Data were analyzed and visualized using Wave 2.3.0 software (Agilent Technologies), and values of OCR and ECAR were normalized to the total protein levels (Bradford Reagent assay, Sigma-Aldrich) in each well. The experiment was performed with three replicates.

## Figures and Tables

**Figure 1 ijms-23-05978-f001:**
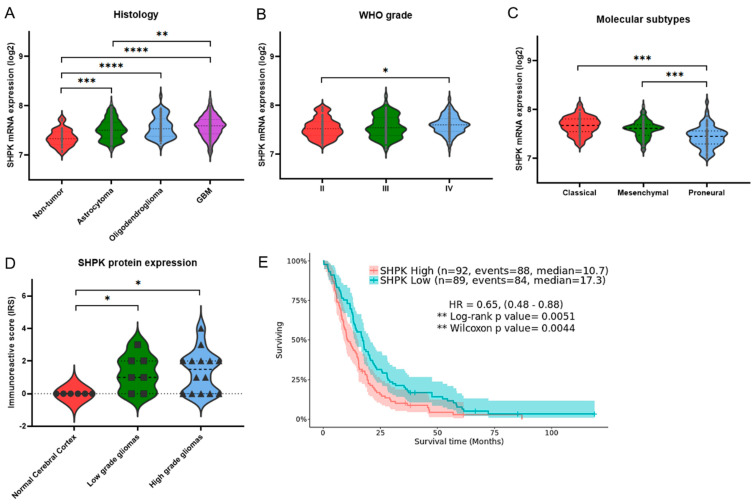
SHPK expression correlates with clinical characteristics and prognosis of GBM patients. (**A**) SHPK mRNA expression within healthy cerebral tissues and different brain tumor histologies. (**B**) SHPK mRNA expression of different WHO-grade brain tumors. (**C**) SHPK mRNA expression in the three different molecular subtypes of brain tumors. (**D**) SHPK protein expression of histological sections from normal and cancer tissues obtained by immunohistochemistry. (**E**) Survival analysis with Kaplan–Meier estimator and visualization of confidence intervals of GBM samples using the median of SHPK mRNA expression values as the cutoff. Hazard Ratio (HR) and *p*-values (Log-Rang and Wilcox) are also shown. Violin plot *p*-values were calculated using an unpaired nonparametric test, the two-tailed Mann–Whitney, with GraphPad Prism 9.3.1. **** *p* < 0.0001; *** *p*< 0.001; ** *p* < 0.01; * *p* < 0.05.

**Figure 3 ijms-23-05978-f003:**
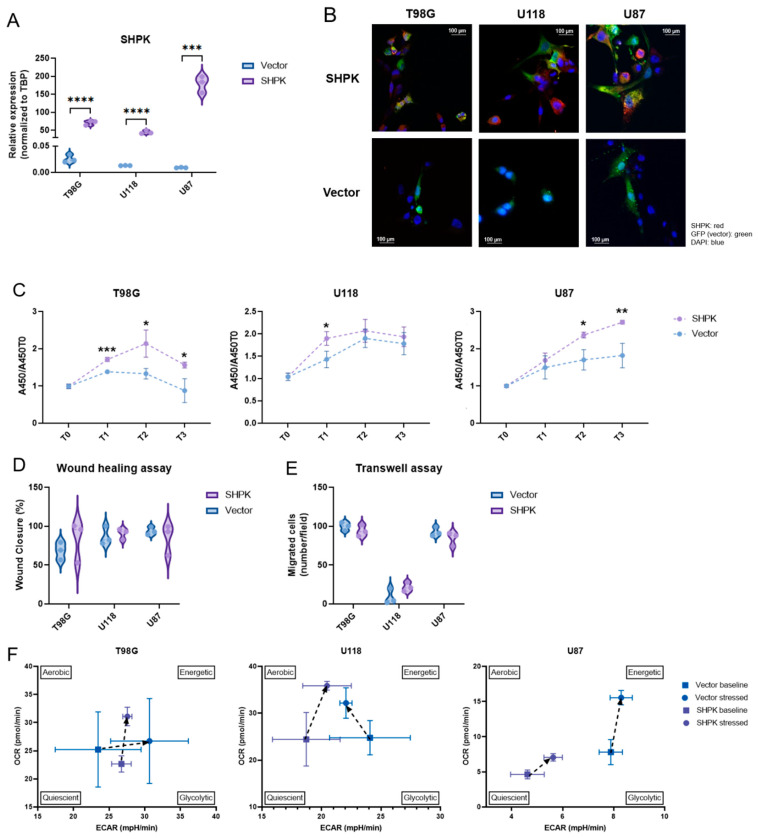
SHPK overexpression and cell functional studies. (**A**) SHPK mRNA expression in T98G, U87, and U118 cells after SHPK overexpression and relative controls. (**B**) Immunofluorescence staining of SHPK protein (red) in T98G, U87, and U118 cells overexpressing SHPK and relative controls (vector alone, green). Nuclei were stained with DAPI (blue). (**C**) Viability of T98G, U87, and U118 cells after SHPK overexpression and their controls at the time of seeding (T0) and after 24 h (T1), 48 h (T2), and 72 h (T3). A450 absorbance values relative to T0 are shown in the vertical axis (*y*). (**D**) Wound healing assay of T98G, U87, and U118 cells after SHPK overexpression and relative controls. (**E**) Transwell migration assay of T98G, U87, and U118 cells after SHPK overexpression and relative controls. Cells that crossed the membrane were counted in five visual fields as migrated cells. (**F**) differences in the metabolic phenotype of T98G, U118, and U87 cells under both basal and stress conditions in the presence or absence of SHPK overexpression. Each measure of OCR and ECAR was calculated by averaging the measurements made in triplicate (SD shown) for three different measurements (9 total measurements) in both the baseline and stressed states. *p*-values were calculated using a two-tailed, unpaired t-test with GraphPad Prism 9.3.1. **** *p* < 0.0001; *** *p* < 0.001; ** *p* < 0.01; * *p* < 0.05.

**Table 1 ijms-23-05978-t001:** Functional enrichment analysis of DEGs.

**abs(LogFC) < 1**	**Reactome**	**ID**	***p*-Value**	**APOD**	**POPDC3**	**CNTN2**	**NKAIN2**	**RUNDC3B**	**KCNQ5**	**RUNDC3A**	**RYR2**	**MYRIP**	**NDN**	**FAM155A**	**FAIM2**	**SPOCK1**	**CHGA**	**PKP4**	**SNCA**	**RAB3A**	**CHGB**	**JPH4**	**TMEFF1**	**NPY1R**	**PAK3**	**DOCK3**	**NAP1L3**	**FA2H**	**XKR4**
	Transmission across Chemical Synapses	112315	8.79 × 10^−13^																										
	Neuronal System	112316	1.15 × 10^−12^																										
	Serotonin Neurotransmitter Release Cycle	181429	3.97 × 10^−8^																										
	Dopamine Neurotransmitter Release Cycle	212676	3.35 × 10^−7^																										
	Neurotransmitter release cycle	112310	7.56 × 10^−6^																										
	Neurotransmitter receptors and postsynaptic signal transmission	112314	1.47 × 10^−5^																										
	Glutamate Neurotransmitter Release Cycle	210500	2.18 × 10^−5^																										
	Acetylcholine Neurotransmitter Release Cycle	264642	1.10 × 10^−4^																										
	Norepinephrine Neurotransmitter Release Cycle	181430	1.54 × 10^−4^																										
	GABA receptor activation	977443	5.65 × 10^−4^																										
	Protein–protein interactions at synapses	6794362	6.42 × 10^−3^																										
	GABA synthesis, release, reuptake and degradation	888590	8.24 × 10^−3^																										
	Long-term potentiation	9620244	1.58 × 10^−2^																										
	Neurotoxicity of clostridium toxins	168799	3.70 × 10^−2^																										
**abs(LogFC) > 1**	**Reactome**	**ID**	***p*-Value**	**PTX3**	**COL1A2**	**IGFBP2**	**ADM**	**CRISPLD1**	**VEGFA**	**COL3A1**	**IGFBP3**	**CDCA7L**																	
	Integrin cell surface interactions	216083	8.21 × 10^−6^																										
	Nonintegrin membrane-ECM interactions	3000171	2.45 × 10^−5^																										
	ECM proteoglycans	3000178	1.16 × 10^−4^																										
	Crosslinking of collagen fibrils	2243919	1.84 × 10^−4^																										
	Scavenging by Class A Receptors	3000480	2.32 × 10^−4^																										
	Collagen formation	1474290	3.23 × 10^−4^																										
	Extracellular matrix organization	1474244	4.40 × 10^−4^																										
	Syndecan interactions	3000170	8.80 × 10^−4^																										
	Assembly of collagen fibrils and other multimeric structures	2022090	9.94 × 10^−4^																										
	Collagen biosynthesis and modifying enzymes	1650814	1.72 × 10^−3^																										
	Collagen chain trimerization	8948216	7.61 × 10^−3^																										
	Anchoring fibril formation	2214320	7.90 × 10^−3^																										
	Collagen degradation	1442490	3.37 × 10^−2^																										

**Table 2 ijms-23-05978-t002:** Gene with mutational status significantly associated with SHPK mRNA expression level.

Gene	Log2FC (Median)	*p*-Value	FDR (BH)	Event_SD	Event_TD
PKHD1L1	−0.8033	0.0304	0.8846	141	3
FZD10	−0.7171	0.0227	0.8846	141	3
IDH1	−0.6885	0.0005	0.2030	141	8
ARMC3	−0.6270	0.0327	0.8846	141	3
SCN9A	−0.6201	0.0052	0.4584	141	5
ATRX	−0.5071	0.0042	0.4584	141	8
ACSM2B	−0.4953	0.0418	0.8846	141	3
PIK3R1	−0.2773	0.0201	0.8846	141	12
TP53	−0.1720	0.0026	0.4584	141	45
EGFR	0.1987	0.0053	0.4584	141	45
RYR2	0.2172	0.0296	0.8846	141	12
MOCS3	0.3278	0.0403	0.8846	141	3
FAM123C	0.3311	0.0490	0.8846	141	4
SLC4A1	0.3325	0.0332	0.8846	141	4
CDH9	0.3434	0.0277	0.8846	141	5
DNAH2	0.3726	0.0419	0.8846	141	5
SPEG	0.3916	0.0234	0.8846	141	4
TRRAP	0.4470	0.0284	0.8846	141	4
MLL2	0.4534	0.0449	0.8846	141	4
XIRP2	0.4578	0.0463	0.8846	141	3
ARHGEF16	0.4721	0.0218	0.8846	141	3
VPS8	0.6664	0.0108	0.7768	141	3

Gene, gene in given target dataset whose association with SHPK expression has been performed. Log2FC (median), change in gene expression level expressed in log2 of mutated/WT ratio. *p*-value, *p*-value obtained from the Wilcoxon statistical test. FDR (BH), false discovery rate calculated by BH (Benjamini–Hochberg method). Event_SD, Number of observations in search dataset attribute without NA’s and Zero’s. Event_TD, Number of observations in target dataset attribute without NA’s and Zero’s.

**Table 3 ijms-23-05978-t003:** The immunoreactive score (IRS).

A (Percentage of Positive Cells)	B (Intensity of Staining)	IRS Score (A × B)
0 = no positive cells	0 = no color reaction	0 = negative
1 = <25% of positive cells	1 = weak reaction	1–2 = mild
2 = 25–75% of positve cells	2 = moderate reaction	3–6 = moderate
3 = >75% of positive cells	3 = intense reaction	7–9 = strong

IRS is calculated as the product of multiplication between the score of the proportion of positive cells (0–4) and the score of the staining intensity (0–3). IRS value ranges between 0 and 9.

## Data Availability

All data generated and analyzed are included in the published article.

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
