# Peer review of "Sedoheptulose Kinase SHPK Expression in Glioblastoma: Emerging Role of the Nonoxidative Pentose Phosphate Pathway in Tumor Proliferation"

_ijms, 2022, doi:10.3390/ijms23115978_

Round 1
Reviewer 1 Report
This manuscript primarily examines publically available databases to make the case that the expression of sedoheptulose kinase (SHPK) in glioblastoma (GBM) is a marker of the activity of the non-oxidative arm pentose phosphate pathway (PPP) and a potential therapeutic target. SHPK mRNA expression obtained from different databases was examined in the context of the histological glioma type, WHO grade, and molecular subtype with significant differences associated with more malignant subtypes. GBM patients separated into two groups by the median SHPK mRNA expression in tumor specimens differed significantly in survival, similarly to the RNA/survival data presented in The Human Protein Atlas. RNA expression data relevant to a variety of processes was then compared for high versus low expressing groups and Pearson correlations between the levels of expression of SHPK and non-oxidative and oxidative PPP –related genes assessed. While it was concluded that “SHPK correlates significantly and positively” with a number of these genes that conclusion is open to discussion as the associations, while significant, can only be considered as weak for some of these (Pearson coefficient > .3). Only one of the correlations approached a moderate association (PGLS, Pearson coefficient 0.46). A limited analysis of SHPK expression and gene mutation was also presented, generally with less than 2 fold differences in the levels of expression based on limited sample numbers. SHPK has been implicated in macrophage polarization to M2-like cells resembling tumor-associated macrophages, is highly expressed in astrocytes and microglia, therefore elevated expression in any of these cell types may contribute to the difference between expression in high versus low grade gliomas. Nevertheless, the focus in the manuscript was on the tumor cells themselves with novel studies of the in vitro behavior of three GBM lines transfected to express high levels of SHPK provided to support the concept that SHPK overexpression in glioma cells drives malignancy. The proliferation of the cells was measured by the WST-1 colorimetric assay, showing relatively minor elevations in cell proliferation/viability. Wound healing and transwell assays are also presented without showing significant differences. These data do not provide clear support for the premise that SHPK expression in glioma cells renders them more malignant. Tumor-associated macrophages, MDSC, and microglia are all as likely to contribute to the metabolic changes that drive glioma malignancy. In conclusion there is some evidence presented in the manuscript that suggests the SHPK upregulation is part of the constellation of changes in the metabolic state as gliomas progress but insufficient support that this is a major driver and mediated by the glioma cells.
Author Response
We would like to thank the reviewers for their careful reading of our manuscript entitled: " Sedoheptulose Kinase SHPK expression in glioblastoma: metabolic effects of pentose phosphate pathway regulation"
We sincerely appreciate all the valuable comments and suggestions, which helped us to improve the quality of the article. Our responses to the reviewers' comment are described in two attached files, each reviewer's comment is shown in bold followed by our response. Appropriate changes, suggested by the Reviewers, have been introduced into the manuscript (highlighted within the paper using the “track changes" system).
We feel that this has resulted in a stronger manuscript and we hope the manuscript will now be suitable for publication in International Journal of Molecular Sciences.

Reviewer 2 Report
The article is important because of the GBM problem is far from being solved. The studing enzyme deserves attention but further complex researches are needed.

Author Response

(The authors gave the same response as above.)

Round 2
Reviewer 1 Report
Data available from such sources as TCGA and the Protein Atlas has been harvested to support the hypothesis that SHPK expression may be a target for therapeutic intervention in glioblastoma. SHPK expression is examined from several aspects and, as can be seen in the various data sources directly, is shown as elevated in more aggressive tumor types. Illustration of this is useful. However the second half of the title "metabolic effects of pentose phosphate pathway regulation" is not as well served. What we are presented is a survey of the relationships between SHPK and PPP gene expression in GBM tissues with generally weak Pearson coefficients even where statistically significant. The data presented certainly do not reflect the "metabolic effects of pentose phosphate pathway regulation" but a survey of the relationships between SHPK and PPP gene expression in GBM tissues. Separation of the samples tested into SHPK high and low groups provides some insight into differences in the patterns of gene expression between these cohorts, which is not unexpected. However, given the range of SHPK expression within different subsets of gliomas it would have been interesting to see if these patterns are consistent within an individual glioma subset. It is also difficult to understand whether the effects of the engineered overexpression of SHPK on glioma cell lines in vitro has any relevance to the disease state as this poorly models the complexity of the glioma tissue where the relationship between cells expressing elevated SHPK and PPP genes remains to be elucidated.
Author Response
We would like to thank the Reviewer for his/her further comments. Each reviewer's comment is indicated in bold followed by our response. Appropriate changes, suggested by the reviewers, have been introduced into the manuscript (highlighted within the document using the "track changes" system).
Data available from such sources as TCGA and the Protein Atlas has been harvested to support the hypothesis that SHPK expression may be a target for therapeutic intervention in glioblastoma. SHPK expression is examined from several aspects and, as can be seen in the various data sources directly, is shown as elevated in more aggressive tumor types. Illustration of this is useful. However the second half of the title "metabolic effects of pentose phosphate pathway regulation" is not as well served. What we are presented is a survey of the relationships between SHPK and PPP gene expression in GBM tissues with generally weak Pearson coefficients even where statistically significant. The data presented certainly do not reflect the "metabolic effects of pentose phosphate pathway regulation" but a survey of the relationships between SHPK and PPP gene expression in GBM tissues.
We realize that the title may actually seem to go beyond what we have achieved, we plan to change it to something more precise and defined. If the reviewer agrees we would propose "Sedoheptulose Kinase SHPK expression in glioblastoma: emerging role of the non-oxidative pentose phosphate pathway in tumor proliferation." We think this title would better describe the work and the results obtained.
Separation of the samples tested into SHPK high and low groups provides some insight into differences in the patterns of gene expression between these cohorts, which is not unexpected. However, given the range of SHPK expression within different subsets of gliomas it would have been interesting to see if these patterns are consistent within an individual glioma subset.
It would definitely be of interest to investigate how the role of SHPK is conserved in different subpopulations of GBM, but this implies analyzing it in other, perhaps larger, tumor cohort sizes. Within each subpopulation in fact there are no significant gene numbers to run a functional enrichment analysis. Additionally, evaluation of whether changes in SHPK mRNA expression within different ranges could lead to enrichment of the same pathways would be interesting. In fact, molecular subtypes are already characterized by different ranges of SHPK mRNA expression values. This could also lead to whether there is an expression "cut-off" that can associate with the activation of specific pathways.
It is also difficult to understand whether the effects of the engineered overexpression of SHPK on glioma cell lines in vitro has any relevance to the disease state as this poorly models the complexity of the glioma tissue where the relationship between cells expressing elevated SHPK and PPP genes remains to be elucidated.
Although commercial tumor cell lines represent the simplest tumor model, this continues to represent the most widely used and a necessary step in the study of biomarkers and anticancer treatments. SHPK overexpression significantly increases proliferation in 3 different glioblastoma cell lines, and this certainly represents a consistent result and is in line with the literature. In addition to this, its overexpression also did not affect migration and invasion, which, again in agreement with the literature, is not correlated with PPP as proliferation, but with glycolysis. This increased proliferation was demonstrated, in this study, in tumor astrocytes, but we know that when we talk about tumor progression, it is never due to one gene or one cellular compartment of the microenvironment. What we would like to bring with this work is a starting point for further studies from both cellular and molecular perspectives. Definitely, studying how SHPK expression in tumor astrocytes influences and is influenced by the tumor microenvironment, as well as understanding which other cell subtypes undergo variations in expression, is something worth investigating based on the findings of our study. Of course, another key aspect would be to investigate biochemically the non-ox pathway of PPP and perform enzyme assays and characterizations of substrates and reaction products. We added this issue to the conclusions of the discussion.
Round 3
Reviewer 1 Report
While the limitations of sample availability restrict more detailed analysis of SHPK and PPP gene expression in glioblastoma the relatively weak correlations shown here provide insight into more general differences in the metabolic states of glioma subsets. The data provided supports the concept that SHPK may be one of the drivers of these metabolic differences and the title change stresses that understanding the relationship with elements of the PPPs is a work in progress, as appropriate.